# ONLINE CONTINUAL LEARNING WITHOUT THE STORAGE CONSTRAINT

## ABSTRACT

Traditional online continual learning (OCL) research has primarily focused on mitigating catastrophic forgetting with fixed and limited storage allocation throughout an agent's lifetime. However, a broad range of real-world applications are primarily constrained by computational costs rather than storage limitations. In this paper, we target such applications, investigating the online continual learning problem under relaxed storage constraints and limited computational budgets. We contribute a simple algorithm, which updates a kNN classifier continually along with a fixed, pretrained feature extractor. We selected this algorithm due to its exceptional suitability for online continual learning. It can adapt to rapidly changing streams, has zero stability gap, operates within tiny computational budgets, has low storage requirements by not storing images but only features, and has a consistency property: It *never forgets* previously seen data. These attributes yield significant improvements, allowing our proposed algorithm to outperform existing methods by over 20% in accuracy on two large-scale OCL datasets: Continual LOCalization (CLOC) with 39M images and 712 classes and Continual Google Landmarks V2 (CGLM) with 580K images and 10,788 classes, even when existing methods use all previously seen images. Furthermore, we achieve a superior performance with considerably reduced computational and storage expenses.

## 1 INTRODUCTION

Online continual learning algorithms need to update a model continuously over a stream of data originating from a non-stationary distribution. This requires them to solve a number of problems: they needs to successfully learn the main task (accuracy), adapt to changes in the distribution (rapid adaptation), and retain information from the past (prevent catastrophic forgetting). A key motif in recent work on online continual learning is the designing algorithms that achieve a good trade-off between these possibly competing objectives under resource constraints (Cai et al., 2021).

To establish the resource constraints for typical commercial settings, we first ask: what is required of continual learning algorithms? A continual learning algorithm must deliver accurate predictions, scale to large datasets encountered during its operational lifetime, and operate within the system's total cost budget (in dollars). The economics of data storage have been studied since 1987 (Gray & Putzolu, 1987; Gray & Graefe, 1997; Graefe, 2009; Appuswamy et al., 2017). Table 1 summarizes the trends, show a rapid decline in storage costs over time ($\sim$ \$100 to store CLOC, the largest dataset for OCL (Cai et al., 2021), in 2017). In contrast, training an ER baseline (Cai et al., 2021), the state-of-the-art OCL method on CLOC currently costs over \$2000 on a GCP server and is still not commercially feasible on mobile devices. Consequently, computational constraints are the primary concern, with storage costs being relatively small. Therefore, as long as computational costs are controlled, economically storing the entire incoming data stream is often feasible.

However, online continual learning has primarily been studied under limited storage constraints (Lopez-Paz & Ranzato, 2017; Chaudhry et al., 2019a; Aljundi et al., 2019b), with learners only allowed to store a subset of incoming data. This constraint has led to many algorithms focusing on identifying a representative data subset (Aljundi et al., 2019b; Yoon et al., 2022; Chrysakis & Moens, 2020; Bang et al., 2021; Sun et al., 2022; Koh et al., 2022). While limited storage aligns with the practical constraints of biological learning agents and offline embodied artificial agents, deep learning-based systems are largely compute-constrained and demand high throughput. Such systems need to process incoming data points faster than the rate of the incoming stream to effectively keep up with the data stream. Cai et al. (2021) shows that even with no storage constraints,

Table 1: The cost of storing data has decreased rapidly, allowing the storage of a large dataset for a negligible cost compared to the cost of computation for both cloud and mobile systems.

| Cloud Storage | 1987 | 1997 | 2007 | 2017 | Mobile NAND | 2000 | 2005 | 2010 | 2017 | 2022 |
|---|---|---|---|---|---|---|---|---|---|---|
| $/MB | 83.33 | 0.22 | 0.0003 | 0.00002 | $/GB | 1100 | 85 | 1.83 | 0.25 | 0.03 |
| Cost of storing CLOC ($) | 350M | 920K | 1250 | 83 | Cost of storing CLOC ($) | 4M | 300K | 6350 | 850 | 70 |
| Training Cost (ER) | | | >2000$ | | Training Cost (ER) | | | Not Currently Feasible | | |

the online continual learning problem is hard as limited computational budgets implicitly limit the set of samples to be used for each training update. Our paper addresses the online continual learning problem with a focus on computational budgets, and not primarily tackling a storage limitations.

Current continual learning approaches aim to encode past seen information implicitly in model parameters. However, encoding new information into a limited, poorly understood parameter space inadvertently causes overwriting past knowledge, known as catastrophic forgetting (McCloskey & Cohen, 1989). In this work, we instead show that fixing feature representations pre-trained and combining this with kNNs, we can reasonably tackle a broad range of complex practical tasks ranging from geolocalization over 39 million images on YFCC datasets (CLOC), and perform long-tailed fine-grained landmark classification (CGLM) despite ImageNet1K features themselves being ill suited for geolocation or fine-grained landmark classification tasks.

Tackling online continual learning with an approximate k-nearest neighbor (kNN) algorithm (Malkov & Yashunin, 2018) over pre-trained features might be perceived as too restrictive. However, we believe it is not only effective but rather future proof. Considering the large-data and large-model trend, we believe current pre-trained features are not only effective, they will get better with time. Moreover, approximate kNN has various desirable features making it highly suitable for online continual learning. Specifically, i) Approximate kNNs are inherently incremental algorithms with explicit insert and retrieve operations allowing it to rapidly adapt to incoming data; ii) With a suitable representation, approximate kNN algorithms are exceptionally effective models at large-scale (Efros, 2017); iii) kNN does not need to store the images itself but low-dimensional features which significantly alleviates the data leakage risk and further reduces storage costs; iv) Despite utilizing all stored features, approximate kNN algorithms are computationally cheap, with a graceful logarithmic scaling of computation; v) kNN will never forget past data. In other words, if a data point from history is queried again, the query yields the same label; vi) It has no stability gap, which is caused by SGD based optimization (De Lange et al., 2022) [1].

Additionally, our approach overcomes a significant limitation of existing gradient-descent-based methods: the ability to learn from a single example. In contrast, updating a deep network for every incoming sample is computationally infeasible. While this does not solve the underlying continual representation learning problem, we show in our work the effectiveness of a simple method on large-scale online continual learning problems, demonstrating viability to many real-world applications.

**Problem formulation.** We formally define the online continual learning (OCL) problem following Cai et al. (2021). In classification settings, we aim to continually learn a function $f : \mathcal{X} \rightarrow \mathcal{Y}$, parameterized by $\theta_t$ at time $t$. OCL is an iterative process where each step consists of a learner receiving information and updating its model. Specifically, at each step $t$ of the interaction,

1. *One* data point $x_t \sim \pi_t$ sampled from a non-stationary distribution $\pi_t$ is revealed.
2. The learner makes the scalar prediction $\hat{y}_t = f(x_t; \theta_t)$ using a compute budget, $B_t^{pred}$.
3. Learner receives the true label $y_t$.
4. Learner updates the model $\theta_{t+1}$ using a compute budget, $B_t^{learn}$

We evaluate the performance using the metrics forward transfer (adaptability) and backward transfer (information retention) as given in Cai et al. (2021). A critical aspect of OCL is the budget in the second and fourth steps, which limits the computation that the learner can expend. A common choice in past work is to impose a fixed limit on storage and computation (Cai et al., 2021). We remove the storage constraint and argue that storing the entirety of the data is cost-effective as long as impact on computation is controlled. We relax the fixed computation constraint to a logarithmic constraint. In other words, we require that the computation time per operation fit within $B_t^{pred}, B_t^{learn} \sim \mathcal{O}(\log t)$. This construction results in total cost scaling $\mathcal{O}(n \log n)$ with the amount of data.

---

[1] We expand the discussion on these properties in detail in Section 3

Table 2: Breakdown of popular OCL systems, with key contributions in red. Most methods focus on sampling techniques for storing datapoints, which cannot transfer here as we store all past samples.

| Works | MemSamp | BatchSamp | Loss | Other Cont. |
|---|---|---|---|---|
| ER (Base) | Random | Random | CEnt | - |
| GSS (Aljundi et al., 2019b) | GSS | Random | CEnt | - |
| MIR (Aljundi et al., 2019a) | Reservoir | MIR | CEnt | - |
| ER-Ring (Chaudhry et al., 2019b) | RingBuf | Random | CEnt | - |
| GDumb (Prabhu et al., 2020) | GreedyBal | Random | CEnt | MR |
| HAL (Chaudhry et al., 2021) | RingBuf | Random | CEnt | HAL |
| CBRS (Chrysakis & Moens, 2020) | CBRS | Weighting | CEnt | - |
| CLIB (Koh et al., 2022) | ImpSamp | Random | CEnt | MR, AdO |
| CoPE (De Lange & Tuytelaars, 2021) | CBRS | Random | PPPLoss | - |
| CLOC (Cai et al., 2021) | FIFO | Random | CEnt | AdO |
| InfoRS (Sun et al., 2022) | InfoRS | Random | CEnt | - |
| OCS (Yoon et al., 2022) | OCS | Random | CEnt | - |
| AML (Caccia et al., 2022) | Reservoir | PosNeg | AML/ACE | - |

## 2    RELATED WORK

**Formulations.** Parisi et al. (2019) and De Lange et al. (2020) have argued for improving the realism of online continual learning benchmarks. Earliest formulations (Lopez-Paz & Ranzato, 2017) worked in a task-incremental setup, assuming access to which subset of classes a test sample is from. Subsequent mainstream formulation (Aljundi et al., 2019b;a) required models to predict across all seen classes at test time, with progress in the train-time sample ordering (Bang et al., 2021; Koh et al., 2022). However, Prabhu et al. (2020) highlighted the limitations of current formulations by achieving good performance despite not using any unstored training data. Latest works (Hu et al., 2022; Cai et al., 2021; Lin et al., 2021) overcome this limitation by testing the capability for rapid adaptation to next incoming sample and eliminate data-ordering requirements by simply using timestamps of real-world data streams. Our work builds on the latest generation of formulation by Cai et al. (2021). Unlike Cai et al. (2021), we perform one-sample learning; in other words, we entirely remove the concept of task by processing the incoming stream one sample at a time, in a truly online manner. Additionally, we further remove the storage constraint which is the key to eliminating degenerate solutions like GDumb (Prabhu et al., 2020).

**Methods.** Traditional methods of adapting to concept drift (Gama et al., 2014) include a variety of approaches based on SVMs (Laskov et al., 2006; Zheng et al., 2013), random forests (Gomes et al., 2017; Ristin et al., 2015; Mourtada et al., 2019), and other models (Oza & Russell, 2001; Mensink et al., 2013). They offer incremental additional and querying properties, most similar to our method, but have not been compared with recent continual learning approaches (Ostapenko et al., 2022; Hayes & Kanan, 2020; Hayes et al., 2019). We perform extensive comparisons with them.

The (online) continual learning methods designed for deep networks are typically based on experience replay (Chaudhry et al., 2019b) and change a subset of the three aspects summarized in Table 2: (i) the loss function used for learning, (ii) the algorithm to sample points into the replay buffer, and (iii) the algorithm to sample a batch from the replay buffer. Methods to sample points into the replay buffer include GSS (Aljundi et al., 2019b), RingBuffer (Chaudhry et al., 2019b), class-balanced reservoir (Chrysakis & Moens, 2020), greedy balancing (Prabhu et al., 2020), rainbow memory (Bang et al., 2021), herding (Rebuffi et al., 2017), coreset selection (Yoon et al., 2022), information-theoretic reservoir (Sun et al., 2022), and samplewise importance (Koh et al., 2022). These approaches do not apply to our setting because we simply remove the storage constraint. Approaches to sampling batches from the replay buffer include MIR (Aljundi et al., 2019a), ASER (Shim et al., 2021), and AML (Caccia et al., 2022). These require mining hard negatives or performing additional updates for importance sampling over the stored data, which face scaling issues to large-scale storage as in our work. We compare with some of the above approaches, including ER as proposed in Cai et al. (2021), that finetune the backbone deep network with one gradient update for incoming data, with unrestricted access to past samples for replay.

**Pretrained representations.** Pretrained representations (Yuan et al., 2021; Caron et al., 2021; Chen et al., 2021; Ali et al., 2021) have been utilized as initializations for continual learning, but in settings with harsh constraints on memory (Wu et al., 2022; Ostapenko et al., 2022). Inspired by Ostapenko et al. (2022), we additionally explore suitability of different pretrained representations. Another emerging direction for using pretrained models in continual learning has been prompt-tuning as it produces accurate classifiers while being computationally efficient (Wang et al., 2022b;a; Chen et al.,

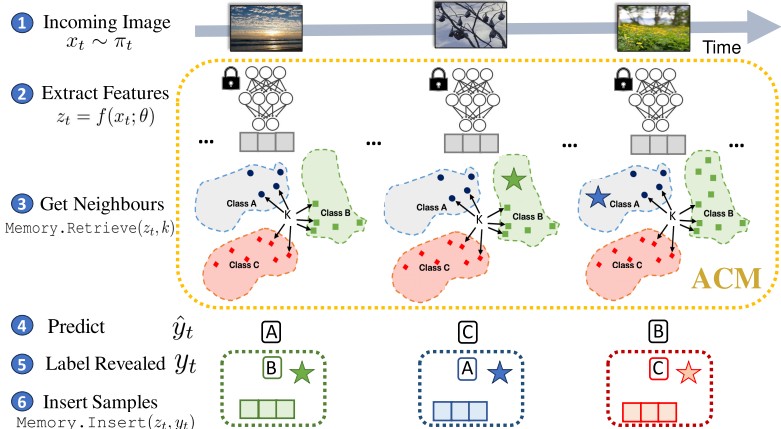

Figure 1: Adaptive Continual Memory (ACM) performs `Memory.Retrieve` and `Memory.Insert` on new incoming samples, extracted by a fixed, pretrained deep network.

2023). However, Janson et al. (2022) show that simple NCM classification outperforms complex prompt tuning strategies. Lastly, the direction most similar to ours is methods which use kNN classifiers alongside deep networks for classification (Nakata et al., 2022; Iscen et al., 2022). We operate in significantly different setting and constraints, use far weaker pretrained representations (ImageNet1K) and benchmark on far larger online classification datasets.

## 3 OUR APPROACH: ADAPTIVE CONTINUAL MEMORY

We use pre-trained feature representations and only learn using the approximate k-nearest neighbor algorithm. Hence, our algorithm is rather simple. We refer to our algorithm as Adaptive Continual Memory (ACM) and refer to the kNN neighbour set as Memory. At each time step, our continual learner performs the following steps:

1. *One* data point $x_t \sim \pi_t$ sampled from a non-stationary distribution $\pi_t$ is revealed.
2. Learner extracts features $z_t = f(x_t; \theta)$
3. Learner retrieves nearest neighbors $\mathcal{N}_t = $ `Memory.Retrieve`$(z_t, k)$.
4. Learner makes the prediction $\hat{y}_t = $ `majority-vote`$(\mathcal{N}_t)$.
5. Learner receives the true label $y_t$.
6. Learner inserts new data: `Memory.Insert`$(z_t, y_t)$.

We summarize this approach in Figure 1. Before presenting further implementation details, we discuss four properties of this method in detail.

**Fast adaptation.** Suppose the learner makes a mistake in a given time step. If the same data point is received in the next time step, the learner will produce the correct answer. By leveraging nearest neighbors, we enable the system to incorporate new data immediately and locally modify its answers in response to as little as a single datapoint. Such fast adaptation, a core desideratum in online continual learning, is infeasible with gradient descent strategies and is not presently a characteristic of deep continual learning systems.

**Consistency.** Consider a hypothetical scenario in which a data point is queried at multiple time instances. Our learner will never forget the correct label for this data point and will consistently produce it when queried, even after long time spans. While learning and memory are much more general than rote memorization, producing the correct answer on previously seen data is an informative sanity check. For comparison, continual learning on deep networks forgets a large fraction of previously seen datapoints even with a minimal delay (Toneva et al., 2019).

**Zero Stability Gap.** When learning new points, traditional continual learning algorithms first drop in performance on past samples due to a large drift from current minima and gradually recover performance on convergence. This phenomena is called as the stability gap (De Lange et al., 2022). Approximate kNN inherently does not have SGD based optimization, instead has pure indexing operations hence inherently enjoys a zero stability gap.

**Efficient Online Hyperparameter Optimization.** Hyperparameter optimization is a critical issue during the online continual learning phase because, as distributions shifts, hyperparameters must be recalibrated. Selecting hyperparameters relevant to optimization like learning rate and batch size, can be nuanced; an incorrect choice has the potential to indefinitely impede future performance. Common strategies include executing multiple simultaneous online learning tasks using diverse parameters (Cai et al., 2021). However, this can be prohibitively resource-intensive. In contrast, our method has a single hyperparameter ($k$), which only affects the immediate prediction, and can be recalibrated during the online continual learning phase with minimal computational cost. We do this by first retrieving the 512 nearest neighbours in a sorted order and subsequently searching over smaller $k$ in powers of two within this ranked list, and selecting the $k$ which achieves highest accuracy on simulating arrival of previous samples.

## 3.1 COMPUTATIONAL COST AND STORAGE CONSIDERATIONS

In the presented algorithm above for our method, feature extraction (step 2) and prediction (step 4) have a fixed overhead cost. However, nearest-neighbour retrieval (step 3) and inserting new data (step 6) can have high computational costs if done naively. However, literature in approximate k-nearest neighbours (Shakhnarovich et al., 2006) has shown that we can achieve high performance while significantly reducing computational complexity from linear $\mathcal{O}(n)$ to logarithmic $\mathcal{O}(\log n)$, where $n$ is the number of data points in memory. By switching from exact kNN to HNSW-kNN, we reduce the comparisons from 30 million to a few hundred, while maintaining a similar accuracy. We utilize the HNSW algorithm from HNSWlib because of its high accuracy, approximate guarantees and practically fast runtime on ANN Benchmarks (Aumüller et al., 2020). We perform a wall-clock time analysis quantifying this speed in Section 4.3.

## 4 EXPERIMENTS

We first describe our experimental setup below and then provide comprehensive comparisons of our method against existing incremental learning approaches.

**Datasets.** We used a subset of Google Landmarks V2 and YFCC-100M datasets for online image classification. These datasets are ordered by the timestamps of image uploads, and our task is to predict the label of incoming images. We followed the online continual learning (OCL) protocol as described in Chaudhry et al. (2019a): We first tune hyperparameters of all OCL algorithms on a pretraining set, continually train the methods on the online training set while measuring rapid adaptation performance, and finally evaluated information retention on a unseen test set. Further dataset details are available in the Appendix.

**Metrics.** We follow Cai et al. (2021), measuring average online accuracy until the current timestep $t$ ($a_t$) as a metric for measuring rapid adaptation, given by $a_t = 1/t \sum_{i=1}^{t} \mathbb{1}_{y_i = \hat{y}_i}$ where $\mathbb{1}_{(\cdot)}$ is the indicator function. We additionally measure information retention, i.e. mitigating catastrophic forgetting, after online training on unseen samples from a test set. Formally, information retention for $h$ timesteps ($IR_h$) at time $T$, is defined as $IR_h = 1/h \sum_{t=T-h}^{T} \mathbb{1}_{y_t = \hat{y}_t}$.

**Computational Budget and Pretraining.** To ensure fairness among compared methods, we restrict the computational budget for all methods to one gradient update using the naive ER method. All methods were allowed to access all past samples with no storage restrictions. All methods started with a pretrained ResNet50 model on the ImageNet1K dataset for fairness. Note that we select the ImageNet1K pretrained ResNet50 because despite being a good initialization, is not sufficient by itself to perform well on the selected continual learning benchmarks. We select a fine grained landmark recognition benchmark over 10788 categories (CGLM), and a harder geolocalization task over a far larger dataset of 39 million samples (CLOC).

**OCL Approaches.** We compared five popular OCL approaches as described in (Ghunaim et al., 2023) on the CLOC dataset. For CGLM, we compare among the top two performing methods from CLOC. We provide a brief summary of the approaches:

1. *ER* (Cai et al., 2021): We use vanilla ER without PoLRS and ADRep (Cai et al., 2021) as they did not improve performance.
2. *MIR* (Aljundi et al., 2019a): It additionally uses MIR as the selection mechanism for choosing samples for training (in a task-free manner).

3. *ACE* (Caccia et al., 2022): ACE loss is used instead of cross entropy to reduce class interference.
4. *LwF* (Li & Hoiem, 2017): It adds a distillation loss to promote information retention.
5. *RWalk* (Chaudhry et al., 2018): This method adds a regularization term based on Fisher information matrix and optimization-path based importance scores. We treat each incoming batch of samples as a new task.

*Training Details for Baselines.* CGLM has the same optimal hyperparameters for CLOC. The ResNet50 model was continually updated using the hyperparameters outlined in (Ghunaim et al., 2023). We used a batch size of 64 for CGLM and 128 for CLOC to control the computational costs. Predictions are made on the next batch of 64/128 samples for CGLM/CLOC dataset respectively using the latest model. The model uses a batch size of 128/256 respectively for training, with the remaining batch used for replaying samples from storage.

**Fixed Feature Extractor based Approaches.** In this section, we ablate capabilities specifically contributed by kNN in ACM compared to other continual learning methods which use a common fixed feature extractor. However, the compared baselines do not have the consistency property provided by ACM, ablating the contribution of this property. We use a 2 layer embedder MLP to project the 2048 dimensional features of ResNet to 256 dimensions using the pretrain set. This adapts the pretrained features to domain of the tested dataset, while providing compact storage and increasing processing speed. All below methods operate on these fixed 256 dimensional features for fairness and operate on features normalized by an online scaler for best performance, with one sample incoming at a timestep. Note that the full model continual learning methods did not benefit significantly by this additional adaptation step. We detail the approaches below:

1. *Nearest Class Mean (NCM)* (Mensink et al., 2013; **?**; Janson et al., 2022): This method maintains a mean feature for each class and classifies new samples by measuring cosine similarity with the mean feature.
2. *Streaming LDA (SLDA) (Hayes et al., 2019)*: This is the current state-of-the-art online continual method using fixed-feature extractors. We use the code provided by the authors with $1e - 4$ being optimal shrinkage parameter.
3. *Incremental Logistic Classification* (Tsai et al., 2014): We include traditional incremental logistic classification. We use scikit-learn `SGDClassifier` with Logistic loss.
4. *Incremental SVM* (Laskov et al., 2006): We include traditional online support vector classification. We use scikit-learn `SGDClassifier` with Hinge loss.
5. *Adaptive Random Forests (ARF)* (Gomes et al., 2017): We chose the best performing method from benchmarks provided by the River library [2] called Adaptive Random Forests.
6. *Eigen Memory Trees (EMT)* (Rucker et al., 2022): This is the current state-of-the-art incremental learning method using Trees, outperforming Sun et al. (2019) by large margins.

### 4.1 COMPARISON OF ACM WITH ONLINE CONTINUAL LEARNING APPROACHES

**Online adaptation.** We compare the average online accuracy of ACM to state-of-the-art approaches on CGLM and CLOC datasets in Figure 2. We observe that ACM significantly outperforms previous methods, achieving a 35% and 5% higher absolute accuracy margin on CGLM and CLOC, respectively. This improvement is due to the capability of ACM to rapidly learn new information.

**Information retention.** We compare backward transfer of ACM to current state-of-the-art approaches on CGLM and CLOC in Figure 2. We find that ACM preserves past information much better than existing approaches, achieving 20% higher accuracy on both datasets. On the larger CLOC dataset, we discover that existing methods catastrophically forget nearly all past knowledge, while ACM maintains a fairly high cumulative accuracy across past timesteps. This highlights the advantages of the consistency property, allowing perfect recall of past train samples and subsequently, good generalization ability on similar unseen test samples to past data.

Moreover, comparing the performance of methods to those in the *fast stream* from Ghunaim et al. (2023), it becomes evident that removing memory restrictions (from 40,000 samples) did not sub-

---

[2]`https://riverml.xyz/0.19.0/`

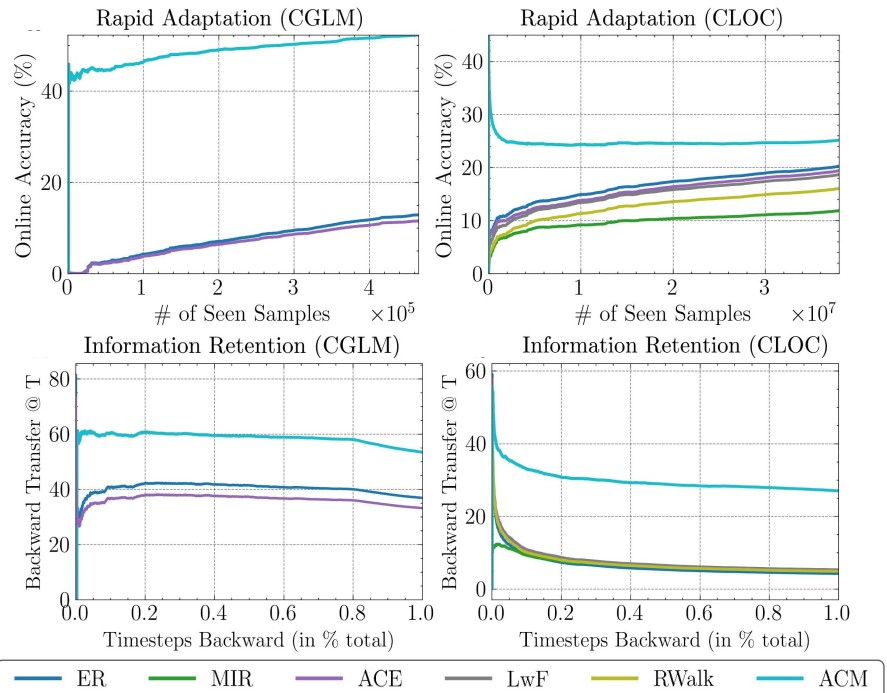

Figure 2: **Online Continual Learning Performance.** We observe that ACM outperforms existing methods by a large margin despite being far cheaper computationally. Traditional methods perform poorly given unrestricted access to past seen data indicating continual learning is a hard problem even without storage constraints. Note that the initial spikes are artifacts of accuracy computation.

stantially alter the performance of traditional OCL methods. This emphasizes that online continual learning with limited computation remains challenging even without storage constraints.

**Key Takeaways.** ACM demonstrates significantly better performance in both rapid adaptation and information retention when compared to popular continual learning algorithms which can update the base deep network on any of the past seen samples with no restrictions. We additionally highlight that ACM has a substantially lesser computational cost compared to traditional OCL methods.

### 4.2    COMPARISON OF ACM WITH APPROACHES LEVERAGING A FIXED BACKBONE

**Online adaptation.** We compare average online accuracy of ACM against recent continual learning approaches that also employ a fixed feature extractor on CGLM and CLOC datasets in Figure 3. We find that ACM outperforms these alternative approaches by significant margins, achieving 10% and 20% higher absolute accuracy on CGLM and CLOC respectively. All approaches here can rapidly adapt to every incoming sample, achieving higher accuracy than traditional OCL approaches. However, ACM can additionally utilizing past seen samples when necessary. Notably, in CLOC, the best alternative approaches collapse to random performance, highlighting that pretrained feature representations are not sufficient, and the effectiveness of kNN despite its simplicity.

**Information retention.** We compare backward transfer of ACM compared to other fixed-feature based OCL approaches on CGLM and CLOC dataset in Figure 3. We observe that ACM outperforms other apporaches by 20% on both datasets, demonstrating its remarkable ability to preserve past knowledge over time. Even after 39 million online updates on the CLOC, ACM preserves information from the earliest samples. In contrast, existing fixed-feature online continual learning methods collapse to random performance.

**Key Takeaways.** The impressive performance of ACM is evident in both rapid adaptation and information retention even amongst latest approaches which similarly use a fixed feature extractor. This further demonstrates the impact of preserving consistency.

**A Note on Time.** ACM and NCM were the fastest approaches among the compared methods. All other approaches were considerably slower, with a 5 to 100 fold increase in runtime compared with

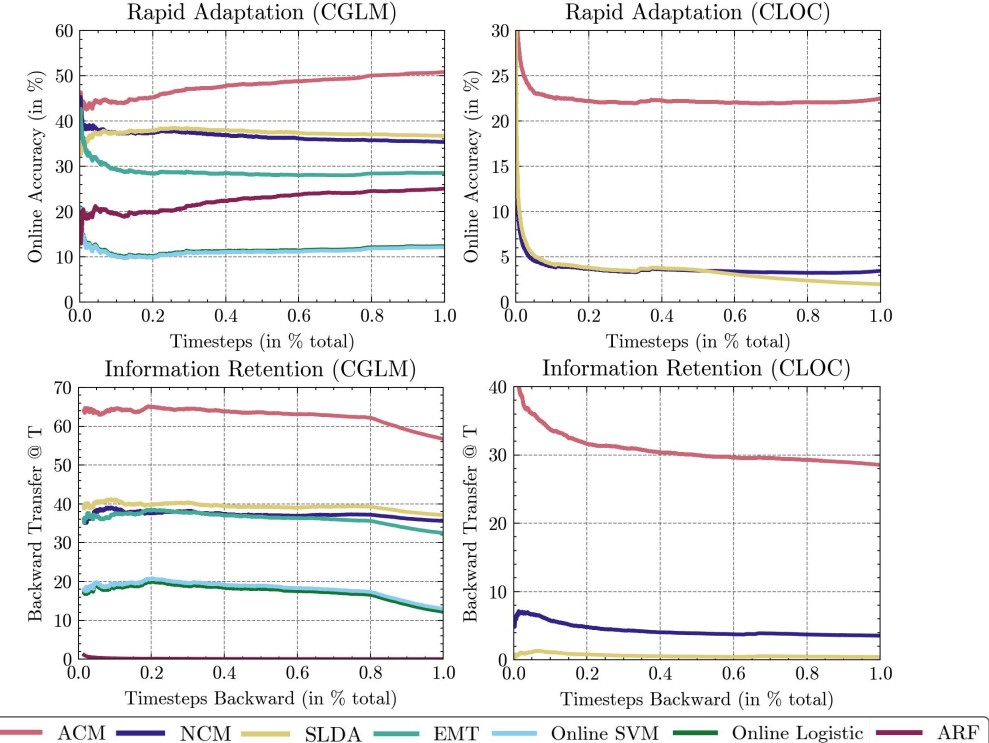

Figure 3: **Online Continual Learning Performance.** ACM outperforms existing methods which leverage a fixed backbone. This highlights the importance of preserving the consistency property in online continual learning. The collapse of other approaches for the CLOC dataset indicates the hardness of the continual learning scenario.

NCM or ACM. However, this could be attributed to codebases, although we used open-source, fast libraries such as River and Scikit-learn.

## 4.3 ANALYZING OUR METHOD: ADAPTIVE CONTINUAL MEMORY

**Contribution of kNN**. Here, we aim to disentangle the benefit provided by the pretrained backbone and the domain tuning by the MLP, with the contribution of kNN to ACM. To test this, we perform online continual learning by replacing the kNN with the fixed MLP classifier. The performance obtained on rapid adaptation on ablating the kNN will be the effect of strong backbone and first session tuning using MLP on pretrain set (Panos et al., 2023). We conducted this experiment using two additional pretrained backbones stronger than our ResNet50 trained on ImageNet1K: A ResNet50 trained on Instagram 1B dataset and the best DINO model XCIT-DINO trained on Imagenet1K to vary the pretraining dataset and architecture.

The results presented in Figure 4 (right). We observe that removing the kNN for classification leads to a drastic decline in performance, indicating that kNN is the primary driver of performance, with performance gains of 20-30%. The decline in performance, losing over 10%, compared to initialization is attributable to distribution shift across time. This is consistently seen across model architectures, indicating that CGLM remains a challenging task with a fixed feature extractor despite backbones far stronger than ResNet50 trained on ImageNet1K.

These findings suggest that kNN is the primary reason for rapid adaptation gains to distribution shifts. Having high-quality feature representations alone or fist-session adaptation is insufficient for a satisfactory online continual learning performance.

**Time Overhead of ACM.** We provide a practical analysis of time to ground the logarithmic computational complexity of ACM. Figure 4 (left) provides insights into the wall-clock time required for the overhead cost imposed by ACM when scaling to datasets of 40 million samples. We observe that the computational overhead while using ACM scales logarithmically, reaching a maximum of approximately 5 milliseconds for 256 dimensional embeddings. In comparison, the time required

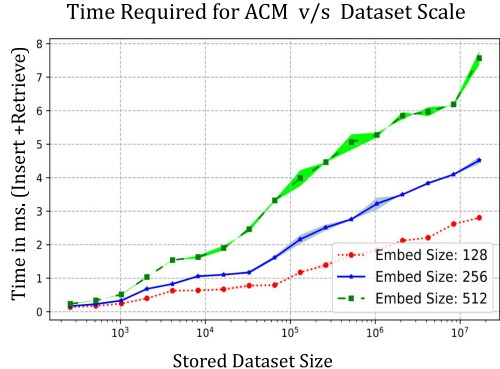 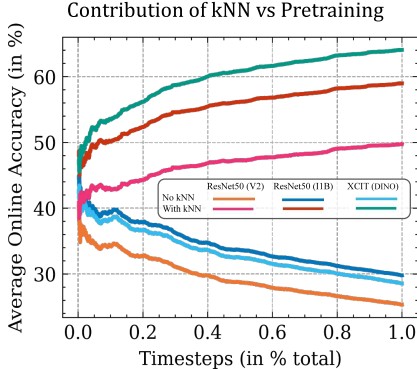

Figure 4: **Left:** Wall clock time overhead of using ACM Memory after feature extraction (x-axis is log-scaled) on a 16-core i7 CPU server. The time increases logarithmically with dataset size, with a 8ms overhead at 40M samples. **Right:** Contribution of kNN and contribution of the original backbone along with the MLP. We observe most of the performance is attributable to the kNN.

for the classification of a single sample for deep models like ResNet50 is approximately 10 milliseconds on an Intel 16-core CPU. It's important to note that when using ACM, the total inference cost of ACM inference would be 15 milliseconds, representing a 50% inference overhead as a tradeoff for high rapid adaptation and information retention performance.

## 4.4 DISCUSSION

**On Fixed Feature Extractor.** A notable limitation of our approach is its dependency on the presence of pretrained features. Consequently, our method may not be appropriate for situations where such features are unavailable. While this limitation is important to acknowledge, it does not diminish the relevance of our approach in situations where pretrained features are available. Our method can be applied effectively in a wide range of visual continual learning scenarios. We have been selective in our choice of models and experiments, opting for pretrained models on ImageNet1K rather than larger models like CLIP or DINOv2 to demonstrate this applicability. Moreover, we tested our approach on more complex datasets like Continual YFCC-100M, which is 39 times larger than ImageNet1K and includes significantly more challenging geolocation tasks.

**On Privacy.** Memory constraints are often linked to privacy concerns. However, it's crucial to understand that merely avoiding data storage does not guarantee privacy in continual learning. Given the tendency of deep neural networks to memorize information, ensuring privacy becomes a much bigger challenge. We highlight that ACM does not store past samples but merely low-dimensional features, making it challenging to reconstruct past data. While a more privacy-preserving adaptation of our method is beyond this paper's scope, one can employ differentially private feature extractors, as suggested by (Yu et al., 2023), to build privacy-preserving ACM models. We conjecture that as more advanced privacy-preserving feature extractors become available, privacy concerns can be addressed in parallel.

**On Mobile.** One interesting aspect for storage constraints are mobile devices as studied in Table 1. Although storage is typically more constrained in mobile devices, we argue computation is far heavily constrained. We highlight that training deep models, as done in continual learning literature is not yet possible on mobile devices. However, since ACM only requires inference in deep networks, it is a feasible solution for mobile-based applications.

**On Real-world Operational Costs.** Finally, to contextualize the computational budget with tangible figures, we envision a hypothetical system necessitating real-time operation on a video stream, facilitated by a 16-core i7 CPU server. Given a feature size of 256 and drawing insights from Figure 4, our method is projected to sustain real-time processing at 30 frames per second for an impressive span of up to 71 years, without necessitating further optimization. Such a configuration would consume approximately 900 GB of storage annually, translating to a cost of roughly $20 per year, as indicated in Table 1. Thus, the ACM stands out as practical, even for the prolonged deployment of continual learning systems.

## 5    CONCLUSION

In this work, we explored online continual learning without any storage constraints. Our reformulation emerges from an analysis of both economic and computational attributes of computing systems. We introduce an approximate kNN-based approach that stores the entirety of the data, adapts on a per-sample basis at each timestep, and still retains computational efficiency. Upon evaluation on large-scale OCL benchmarks, our system yields significant improvements over existing methods. Our approach is computationally cheap and scales gracefully to large-scale datasets.

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

## A    IMPLEMENTATION DETAILS

In this section, we describe the experimental setup in detail including dataset creation, methods and their training details.

### A.1    DATASET DETAILS

**Continual Google Landmarks V2 (CGLM)** (Weyand et al., 2020): We introduce a new dataset which is a subset of Google Landmarks V2 dataset (Weyand et al., 2020) as our second benchmark. We use the train-clean subset, filtering it further based on the availability of upload timestamps on Flickr. We filter out the classes that have less than 25 samples. We uniformly in random sample 10% of data for testing and then use the first 20% of the remaining data as a hyperparameter tuning set, similar to CLOC. We get $430K$ images for continual learning with 10788 classes.

We start with the train-clean subset of the GLDv2 available from the dataset website[3]. We apply the following preprocessing steps in order:

1. Filter out images which do not have timestamp metadata available.
2. Remove images of classes that have less than 25 samples in total
3. Order data by timestamp

We get the rest $580K$ images for continual learning over 10788 classes with large class-imbalance alongside the rapid distribution shift temporally. We allocate the first 20% of the dataset timestamp-wise for pretraining and randomly sample 10% of data from across time for testing.

We provide the scripts for cleaning the dataset alongside in the codebase.

**Continual YFCC-100M (CYFCC)** (Thomee et al., 2016): The subset of YFCC100M, which has date and time annotations (Cai et al., 2021). We follow their dataset splits. We order the images by timestep and iterate over 39 million online timesteps, one image at a time, with evaluation on the next image in the stream. Note that in contrast, Cai et al. (2021) uses a more restricted protocol assuming 256 images per timestep and evaluates on images uploaded by a different user in the next batch of samples. We download the images and the metadata as given by Cai et al. (2021) from their github repository. We provide a guide for downloading alongside in our codebase.

### A.2    MODEL AND OPTIMIZATION DETAILS

**Training OCL approaches**: We use a ResNet50-V2 model from Pytorch for all other methods. We trained models starting from a pretrained ImageNet1K model with a batch size of 128 for CGLM and 256 for CLOC with a constant lr of 5e-3, SGD optimizer as specified in the original work. We used a 80GB A100 GPU server for the training.

**MLP Details**: We use a 2-layer MLP consisting of (input, embedding) and (embedding, output) layers, with batchnorm and ReLU activations trained for 10 epochs on CGLM pretrain set and 2 epochs on the CLOC pretrain set. We do not do hyperparameter optimization and use the default parameters as hyperparameter optimization in online continual learning is an open problem. All ACM experiments were performed on one 48 GB RTX 6000 to extract features and the kNN computation was done on a 12th Gen Intel i7-12700 server.

---

[3]https://github.com/cvdfoundation/google-landmark

# B   EVALUATING RAPID ADAPTATION USING NEAR-FUTURE ACCURACY

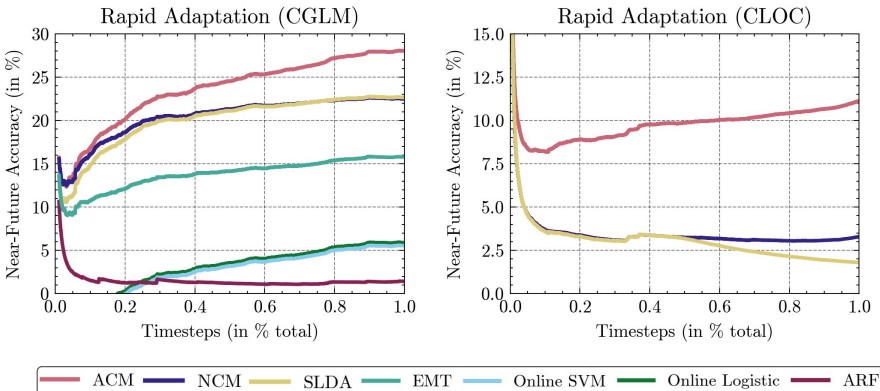

Figure 5: **Online Continual Learning Performance.** We observe that ACM outperforms existing methods on the near-future accuracy metric by 5-10%, maintaining state-of-the-art performance.

**Evaluation of Rapid Adaptation using Near-Future Accuracy.** Hammoud et al. (2023) discovered label correlations in the original online accuracy metric used for evaluating online continual learning methods. However, ACM remained state-of-the-art performance on their *fast stream* setting, despite performance of other methods falling below offline learning baselines.

In this section, we compare the performance of ACM to other online learning approaches with a fixed feature extractor on CGLM and CLOC datasets with the near-future accuracy metric using the same delay parameters (Hammoud et al., 2023). We present our results in Figure 5. We observe that ACM achieves state-of-the-art performance among online continual learning approaches, outperforming them by 5-10% margins. This provides us with additional evidence to indicate that ACM did not exploit on label correlations to achieve high online accuracy.

# C  ADDITIONAL EXPERIMENTS

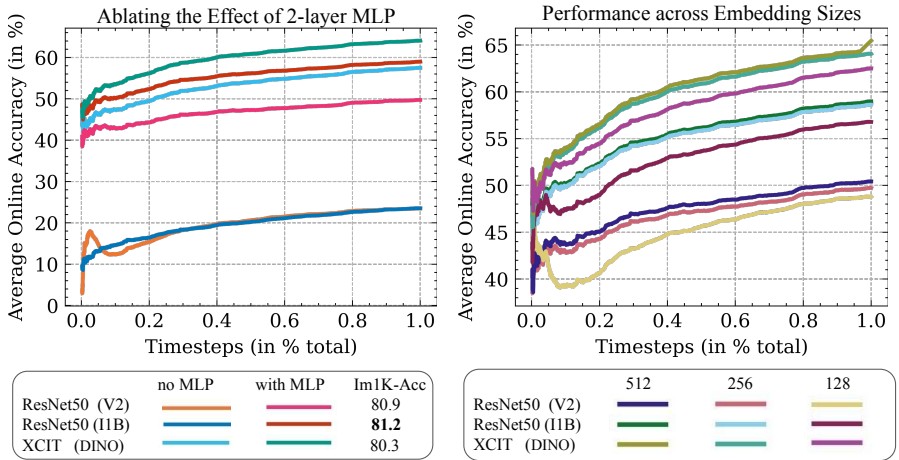

Figure 6: **Left: Ablating the effect of first-session adaptation using an MLP.** We observed first-session adaptation caused minor improvement in performance on XCIT model with 512 dimensional features, however provided a major performance boost for 2048 dimensional ResNet50 features. **Right: Ablating the effect of embedding dimensions.** We observe that varying the embedding size allows a tradeoff between computational cost and accuracy. However, it does not cause dramatic increase or decrease in performance.

**Ablating the contribution of the 2-layer MLP.** We ablate the contribution of first-session adaptation (Panos et al., 2023) on the pretraining set. We compare online learning performance with and without using the MLP and present the results across different models in Figure 6 (left). We observe that the performance in XCIT DINO model achieves a performance boost of 5% when using the MLP, indicating that first-session adaptation allows for an improvement in performance. However, both ResNet50 models gain large improvements of over 30% in online accuracy due to the MLP. We notice that this is attributable to the curse of dimensionality. ResNet50 architecture has a far larger feature dimension of 2048 in compared to XCIT-DINO having 512 dimensions. Comparing models with MLP, we surprisingly observe that ResNet50-I1B performs worse than XCIT-DINO model despite ResNet50-I1B achieving better performance across traditional benchmarks, indicative of robust and generalizable features. We conclude that ResNet50 architecture is a poor fit for ACM due to 2048 dimensional features.

*Conclusion.* Models with high-dimensional embeddings perform much more poorly in combination with a kNN despite better representational power due to the curse of dimensionality.

**Ablating the effect of the embedding size.** We now know that embedding dimensions can significantly affect performance. We vary the embedding dimensions to explore the sensitivity of ACM to embedding dimensions. We start with 512 dimensional XCIT feature dimensions, varying the embedding sizes to 512, 256 and 128.

We present the results in Figure 6 (right). First, we observe that decreasing the embedding dimension to 256 results in minimal drop in accuracy across all the three models but reduces the computational costs by half as shown in Figure 4 in the main paper. Further reduction in embedding dimension leads to considerable loss of performance. An embedding size of 256 achieves the best tradeoff between speed and accuracy.

