# OpenReview forum: "Online Continual Learning Without the Storage Constraint"
_ICLR.cc/2024/Conference — ICLR 2024 Conference Withdrawn Submission_

### Official Review · Reviewer_HFxe · 2023-10-24

**Soundness:** 3 good
**Presentation:** 3 good
**Contribution:** 2 fair
**Rating:** 5
**Confidence:** 5

**Summary:**

In this study, the authors delve into online continual learning under the constraints of limited computational resources, while relaxing storage capacity constrain. The central concept revolves around the integration of a K-nearest neighbors (KNN) approach with a static, pre-trained feature extractor. The authors substantiate the appropriateness of KNN for this scenario, underscoring its capability to swiftly adapt to dynamic data streams without compromising stability. Furthermore, they underscore its efficiency in the context of constrained computational resources, attributed to its ability to store essential features exclusively and uphold a consistency property, thereby preserving previously encountered data.

**Strengths:**

In this paper, online  continual learning in the presence of drift is studies, which indeed is an interesting and a practical topic as data streaming applications keep on increasing. The paper is well written and easy to read, and the algorithm is clearly presented and also demonstrated in Figure 1. The paper very well included state of the art. The improvement obtained in the experiments is considerable.

**Weaknesses:**

In my opinion, the assumed setup seems simplified and unrealistic, given that it presumes the use of a fixed pre-trained feature extraction method for all forthcoming data in the data stream, as also mentioned by the authors. In a data stream, the data distribution and their features structure can evolve, and new classes can emerge, using a fixed pre-trained feature extraction method might not be enough in data streaming learning. The authors discussed about mobile devices, but can one really claim that we don't have  storage constraints in mobile devices in data explosion era?
This core idea in this study is very similar to (Nakata et al., 2022), in both studies KNN has been used as a reminder assistance for the backbone model for quick adaptation after drift. The idea is indeed interesting, but unfortunately it does not contain novelty. I cannot find any major novel differentiation within this work in compare with the mentioned study.

**Questions:**

Can you provide some explanation and comparison with KNN for the approaches mentioned in:"Fixed Feature Extractor based Approaches"
Can you add some justification or empirical results to prove this: "We additionally highlight that ACM has a substantially lesser computational cost compared to traditional OCL methods."

You have used a fixed set of hyperparameters determined from the pretrained set, but did (can) you explore how the need for hyperparameters tuning can vary when observing drift? BTW,  Please mention in the text the concrete hyperparameters here as well: "We first tune hyperparameters of all OCL algorithms on a pretraining set"

Could you please provide experiments or explain how your algorithm in compare with other approaches work when a new class emerges that has not been see in the pretraining phase?

In KNN have you assumed a fixed K?

---

> ### Author Response · Authors · 2023-11-16
> **Rebuttal**
>
> Thank you for the encouraging feedback. We appreciate you finding our time-incremental study practical and impactful, our paper well written and experiments showing clear state-of-the-art performance by considerable margins! We now organize the concerns into points and address them in detail below:
>
> **[W1 Not novel, fixed feature extractors not enough].** We understand the reviewers concern and had pre-emptively addressed this in our discussion section. We re-highlight that:
>
> - Ideally, a continual learner should be able to accumulate new concepts or knowledge in features over time.
> - However, what we show that at least for large scale vision tasks, a rich embedding space generated by a small pretrained model (Imagenet1k) is sufficient enough to perform well on large scale classification tasks (datasets of 39M samples), on a far harder geolocation and fine-grained landmark retrieval task, in an online continual manner.
> - It does show the effectiveness of a simple method on a set of CL problems that are large enough to be of importance to many real-world applications.
> - We understand the sentiment behind the question. Hence, we chose our experiments carefully. Specifically, we chose to use pretrained models on ImageNet1K instead of the larger CLIP or DINOv2 models popular in continual prompt-tuning literature, and chose to test on significantly more complex datasets like Continual YFCC-100M with 39x times the size and extreme data distribution shifts etc as highlighted by the reviewer.
> - We show with comparing with other fixed feature extractor baselines that the problem of generalizing to CGLM and CLOC is very hard. Other state-of-the-art methods with fixed backbones have subpar performance and completely fail to adapt on CGLM and CLOC datasets respectively.
>
> **W2 [Mobile Devices].** We first clarify that we do not argue for absolutely no storage constraint in the paper but merely far looser restrictions on storage backed by concrete statistics in Table 1 *compared to computational costs*. Note that mobiles have prohibitively limited compute and back-propagating in deep nets is ill-suited for mobiles, needing study. In contrast, ACM only has  inference.
> - We however understand the reviewers concern that storage costs can grow, hence we clarify that did heavily optimized the storage requirement for our ACM method.
> - We do not store samples in ACM to minimize storage overhead and overcome other problems with storing past data (other methods however store data). This dramatically reduces the storage cost of YFCC from over 2TB to 70GB for 39 million samples. This requires cost of < $0.1.
>
> **W3 [Nakata et al]. and Novelty** Thanks for the great question! We clarify that Nakata et al.2022 uses (i) exact kNN, (ii) unfairly limits memory of comparisons, critical for outperforming existing methods and (iii) solves relatively simple tasks with powerful pretrained models where kNN is an overkill (W1 of the reviewer).
>
> We argue:
> - We cannot compare with Nakata et al as they use exact kNNs which have O(N^2) evaluation complexity and are prohibitively expensive to scale given we have 39 million evaluation steps. Note that they explicitly list efficient kNN approximation methods as future interesting direction, which is what our work does, having O(N logN) evaluation complexity.
> - Nakata et al.2022 unfairly compares kNN which stores all 1.2M samples with continual learning methods which store only 20K exemplars – a fair comparison must have all approaches storing the same number of exemplars. This allows them to outperform existing methods! In contrast, we allow all methods access to all exemplars and equalize computation budget in Figure 2 (Note that ACM has dramatically lower compute requirements, being disadvantaged, compared to others).
> - They use powerful ImageNet21K pretrained models and test on far smaller datasets, which can be efficiently tackled by NCM based classifiers [1] (W1 concern raised by the reviewer). We, in contrast, show that NCM based classifiers completely fail to both alleviate catastrophic forgetting and rapidly adapt in our scenario.
>
> [1] RanPAC: Random Projections and Pre-trained Models for Continual Learning
>
> **Novelty.**
> - Previous works often introduce algorithmic novelty by designing complex continual learning systems; we believe it is critical to show a simple, computationally cheap but powerful baseline which outperforms popular methods by large margins on large-scale datasets (Fig 2 in paper).
> - We show that kNN based pretrained models outperform models which are continually trained for one epoch on the dataset, preventing catastrophic forgetting (high information retention) cheaply which is new and surprising.
> - In essence, the novelty we introduce is primarily in uncovering important shortcomings in literature along with proposing a simple method that can have a major real-world impact due to its simplicity and deployability with no additional training.

---

> ### Author Response · Authors · 2023-11-16
> **Rebuttal [Questions]**
>
> **Q1 [Computational Cost]** Thank you for the great question! We indeed missed to reasonably justify this, and provide a table of runtime comparisons of the top three performing methods with a static backbone for CGLM dataset with kNN as the baseline (1x) below. All methods are run on the same CPU system for fairness:
>
> | Method | Speed |
> |--------|-----------|
> | kNN    | 1x       |
> | NCM    | 4x       |
> | SLDA   | 12x      |
>
> We will add this comparison to the Appendix.
>
> [kNN outperforming NCM in counterintuitive]: Note that large-scale datasets like CGLM have >10800 classes requiring comparisons with every class. In contrast, HNSW-kNN requires far less comparisons to get nearest neighbors. Typically, HNSW-kNN is upto 2x more expensive than NCM but achieves far higher performance justifying this cost.
>
>
>
> **Q2 and Q4 [Hyperparameters and $k$]** Thank you for the great question! We clarify that:
> - We tune the hyperparameters of all OCL algorithms but ACM on the pretraining set, to compare with their best versions.
> - We did not want to optimize our approach as hyperparameters of our approach ($k$ of the k nearest neighbours) can vary with drift as motivated. We optimize $k$ while training online.
>
> **Q3 [Class-incremental setting]** Thank you for the great question!  We clarify that the standard class-incremental setting doesn't fit in our framework as the new samples will be from the same classes as immediate past examples by construction.
> - We do not assume any clear class boundaries and deal with the general continual learning setup. Therefore, rapid adaptation, a critical metric for online CL, can be evaluated.
> - The fact that the methods for regular online-IL and our setting differ significantly is also discussed in [1] showing that classical online learning methods do not perform well under realistic datasets like CGLM and CLOC.
> - We do encounter new samples of long-tailed classes in CGLM dataset with heavy biases in class distributions. We have shown in our CGLM results that our method can perform well across these settings.
>
> [1] Ghunaim et. al., “Real-Time Evaluation in Online Continual Learning: A New Hope”, CVPR 2023
>
> We would be happy to answer any further questions you have. If you do not have any further questions, we hope that you might consider raising your score.

---

### Official Review · Reviewer_9XgP · 2023-10-30

**Soundness:** 3 good
**Presentation:** 3 good
**Contribution:** 3 good
**Rating:** 5
**Confidence:** 4

**Summary:**

The paper proposes a kNN based retrieval of previous sample from the infinite memory for reminding the past information to alleviate forgetting. While the setup could be realistic as the memory cost becomes negligible, even using the efficient version of kNN, the retrieval of relevant sample from the infinite memory is still computationally expensive and reminding previously learned knowledge may not be desirable as large models are arguably much less forgettable about previously learned knowledge. By the help of perfect remind of previously given data, the method improves the classification accuracy significantly over the other methods.

**Strengths:**

- Superior empirical gain over other methods
- Simplicity of the method
- Good empirical setup using CGLM and CLOC datasets

**Weaknesses:**

- The presented setup with infinite memory is arguably realistic online continual learning setup. The infinite memory would eventually prevent forgetting by perfect reminding (by using properly efficient version of kNN retrieval) and the proposed method is not surprising with that. Thus, it is questionable whether the proposed setup and the method is indeed helping us to solve online continual learning for real world deployment or not.
- Method is not well motivated. It is not clear why the kNN let the model adapt to new sample fast and lead to zero stability gap inherently.
- Why the proposed method only has very high initial accuracy in Rapid adaptation plot using CLOC in Fig. 2 (right upper).

**Questions:**

See weaknesses.

---

> ### Author Response · Authors · 2023-11-16
> **Rebuttal**
>
> Thank you for the encouraging feedback. We appreciate the simplicity of our method as a strength, extensive performance improvements, and our baseline ACM directly impactful to industrial applications! We address the concerns raised below:
>
> **W1. [Setup Unrealistic]** We first clarify that we do not claim *infinite memory* in the paper.
> - We do put looser restrictions on memory storage than continual learning works as disk storage is relatively far cheaper than compute, backed by solid evidence in Table 1 (fairly easy to verify by searching up prices of large-capacity HDD and SD cards   online). We argue storing 39 million samples costs <$100 with storage costs as detailed in Table 1 which is very inexpensive with HDDs and SD cards for cloud and mobile applications.
> - We however understand the reviewers concern that storage costs can grow, hence we clarify that did heavily optimized the storage requirement for our ACM method.
> - We do not store samples in ACM to minimize storage overhead and overcome other problems with storing past data (other methods however store data). This dramatically reduces the storage cost of YFCC from 2TB to 70GB for 39 million samples. This requires cost of less than $0.1, minimizing costs.
>
> We hope this alleviates the concern about practicality of the proposed setup.
>
> **W2 [How does kNN adapt fast and have zero stability gap]** Thanks for the great question! We clarify this below:
> - Current OCL methods which update the deep network are not able to adapt/converge to incoming samples in one gradient step: high learning rates prevent convergence resulting in poor performance, low learning rates lead to insufficient adaptation.
> - ACM, in contrast, does not have SGD based optimization but simple kNN indexing which is the core differentiation. We detail the two specific concerns below:
>
> [Rapid Adaptation]: Concretely this means, given a datapoint encountered had wrong classification and it was indexed with the label, if we were to encounter the same datapoint (or very close datapoint in cosine space with the same label) in the very next timestep, kNN will guaranteed to (approximately guaranteed to) correctly classify that datapoint despite failing for the previous one.
>
> [Zero Stability Gap]: Stability gap arises from the fact that gradient based optimization needs time to converge and has poor solutions in the intermediate time. This especially affects online learning due to the one gradient step constraint preventing convergence. kNN has no optimization but simply indexing hence does not decrease in accuracy before increasing again.
>
> We clarified this point in Section 3 of our updated draft. We hope this alleviates the concern about the motivation of our method.
>
> **W3 [Accuracy is only high on earlier stages].** We would like to push back on this claim by the reviewer.
> - We clarify that we even in the last timestep, we outperform competing approaches by over 5% where the progress amongst competing approaches has been on a 0.5-1% margin as seen in Figure 2.
> - Overall, we consistently outperform competing approaches across the long span of 39M timesteps in CLOC being unambiguously state-of-the-art by a wide margin.
>
> We further push back on an important claim made in the summary:
>
> > large models are arguably much less forgettable about previously learned knowledge
>
> - We have shown this claim to not be true in Figure 2. Despite having access to all previous samples, past methods have dramatic forgetting. In-fact, comparing our memory unconstrained setting and the traditional memory constrained studied in (Ghunaim et. al. 2023) we find that removing the memory constraint had surprisingly little impact on reducing forgetting in deep models.
> - This motivates the need to find methods which can computationally efficiently alleviate forgetting. We propose ACM as a strong contender.
>
> We would be happy to answer any further questions you have. If you do not have any further questions, we hope that you might consider raising your score.

---

### Official Review · Reviewer_2LkF · 2023-11-01

**Soundness:** 2 fair
**Presentation:** 3 good
**Contribution:** 2 fair
**Rating:** 3
**Confidence:** 4

**Summary:**

The proposed submission questions whether the popularly enforced storage constraints in online continual learning is really realistic and develop an online continual learning method with no storage constraints, e.g. storing all data points (whether raw or processed). They use an approximate kNN classifier to make predictions using the stored data points, which boosts fewer computation costs in terms of training and predicting using the continually learned model. Evaluation is done on YFCC-100M and Google Landmarks V2 datasets.

**Strengths:**

- The submission poses an interesting question of whether the storage constraint is realistic or not.
- The proposed method is simple and straightforward.
- The paper reads well.

**Weaknesses:**

- The novelty is limited. The methodology itself is an approximate kNN, with little modifications. Additionally, the method merely uses pretrained feature extractors as well, which does not add to technical novelty.
- The consideration of the storage constraint seems a bit uni-dimensional to me. There are other factors than just storage costs that are not taken into account. For instance, the data itself may be volatile, i.e., some data points may be required by law to be deleted upon a set duration.

**Questions:**

- The feature computation part seems to have little difference with regards whether it is used in a regular, static data manner or in the continual learning setup. Can the authors clarify on this?

---

> ### Author Response · Authors · 2023-11-16
> **Rebuttal**
>
> Thank you for the feedback. We appreciate you finding our change of setting to relaxed storage constraint refreshing, and our baseline ACM simple and our paper easy to read! We address the concerns raised in detail below:
>
> **W1 [Novelty].** We respectfully disagree with the reviewer and would like to push back on this claim.
> - We understand applying kNN to a representation space is not new and several domains (majorly self-supervised learning) apply kNNs for downstream classification using pretrained models .
> - Previous works often introduce algorithmic novelty by designing ad-hoc continual learning components; we believe it is critical to show a simple, computationally cheap but powerful baseline which outperforms popular methods by large margins on large-scale datasets (Fig 2 in paper).
> - We show that kNN based pretrained models outperform models which are continually trained for one epoch on the dataset, preventing catastrophic forgetting (high information retention) cheaply which is new and surprising. We are the first to show the efficacy in a realistic online continual learning setting.
>
> In essence, the novelty we introduce is not in the algorithm itself, but primarily in uncovering important shortcomings in literature along with proposing a simple method that can have a major real-world impact due to its simplicity and deployability with no additional training.
>
> We request the reviewer to reconsider and not dismiss our contributions as “not novel”.
>
> **W2 [Storage constraint uni-dimensional].** We agree with the reviewer that there are some situations where privacy can be violated by storing all the images. However, as we state in the Section 4.4 and introduction, there are a large set of applications where this is not the case, but very few continual learning works focus on those setups which we intend to fix.
>
> - We however understand the reviewers concern and clarify that our ACM method does not store the data itself for our method but simply features which significantly alleviates the privacy risk.
> - We re-highlight a point from Section 4.4, that while works argue the need for privacy, there is *no work that can guarantee private continual learning*, as simply not storing past data is not sufficient for preserving privacy, due to the propensity of deep networks to memorize information.
>
> **Q1 [Feature Computation is invariant].** Yes, the feature computation part does not vary! We develop continual learning approaches on a fixed backbone in this work. Hope that clarifies the question.
>
> We would be happy to answer any further questions you have. If you do not have any further questions, we hope that you might consider raising your score.